# Burnout Syndrome in a Military Tertiary Hospital Staff during the COVID-19 Contingency

**DOI:** 10.3390/ijerph19042229

**Published:** 2022-02-16

**Authors:** Adriana Martínez-Cuazitl, Iván Noé Martínez-Salazar, Guadalupe Maza-De La Torre, Jorge Arturo García-Dávila, Edgardo Alonso Montelongo-Mercado, Antonio García-Ruíz, Héctor Faustino Noyola-Villalobos, Mayra Gabriela García-Araiza, Sergio Hernández-Díaz, Dora Luz Villegas-Tapia, Eira Cerda-Reyes, Arleth Sarai Chávez-Velasco, Juan Salvador García-Hernández

**Affiliations:** 1Research Department, Military Central Hospital (Ministry of National Defense of Mexico-SEDENA), Mexico City 11200, Mexico; adyta0@hotmail.com (A.M.-C.); arlethschavezv@hotmail.com (A.S.C.-V.); medicopixelado@gmail.com (J.S.G.-H.); 2Military School of Medicine (Ministry of National Defense of Mexico-SEDENA), Mexico City 11200, Mexico; 3General Direction of Health (Ministry of National Defense of Mexico-SEDENA), Mexico City 11200, Mexico; matg591212_7@hotmail.com; 4General Direction, Military Central Hospital (Ministry of National Defense of Mexico-SEDENA), Mexico City 11200, Mexico; daviorto@hotmail.com; 5General Deputy Direction, Military Central Hospital (Ministry of National Defense of Mexico-SEDENA), Mexico City 11200, Mexico; edgardo_columna@msn.com; 6Technic Deputy Direction, Military Central Hospital (Ministry of National Defense of Mexico-SEDENA), Mexico City 11200, Mexico; antoniogarciaruiz@me.com; 7Medical Deputy Direction, Military Central Hospital (Ministry of National Defense of Mexico-SEDENA), Mexico City 11200, Mexico; hfnoyola@prodigy.net.mx; 8Teaching and Research Area, Military Central Hospital l (Ministry of National Defense of Mexico-SEDENA), Mexico City 11200, Mexico; mggarciaaraiza@gmail.com; 9Medical Area, Military Central Hospital (Ministry of National Defense of Mexico-SEDENA), Mexico City 11200, Mexico; sehedi@yahoo.com.mx; 10Psychology Subsection, Military Central Hospital (Ministry of National Defense of Mexico-SEDENA), Mexico City 11200, Mexico; doluvita@outlook.com; 11Academic Coordination Department, Military Central Hospital (Ministry of National Defense of Mexico-SEDENA), Mexico City 11200, Mexico; arieirace@yahoo.com.mx

**Keywords:** burnout syndrome, COVID-19 pandemic, tertiary hospital, military and civilian personnel

## Abstract

(1) Background: Burnout syndrome (BOS) is defined as a psychological state of physical and mental fatigue associated with work. The COVID-19 pandemic greatly impacted the physical and mental wellbeing of health professionals. The objective of this work was to determine the impact on personnel, monitoring the frequency of BOS throughout the pandemic. (2) Methods: The Maslach Burnout Inventory (MBI) was self-applied in four periods of the pandemic according to sociodemographic and employment characteristics. In this study, all hospital personnel were included; the association of BOS with sex, age, type of participant (civilian or military), military rank and profession was analyzed. (3) Results: The frequency of BOS was 2.4% (start of the pandemic), 7.9% (peak of the first wave), 3.7% (end of the first wave) and 3.6% (peak of the third wave). Emotional exhaustion (EE) was the most affected factor, and the groups most affected were men under 30 years of age, civilians, chiefs and doctors, especially undergraduate medical doctors and specialty resident doctors, and nursing personnel were less affected. (4) Conclusions: The low BOS levels show that the containment measures and military training implemented by the hospital authorities were effective, although the chief personnel were more affected in the first wave. It is probable that this combination allowed the containment of BOS, which was not observed in civilians.

## 1. Introduction

The COVID-19 pandemic had an overwhelming impact on healthcare professionals. Due to changes in healthcare, cessation of services, adaptation of care areas, reassignment of staff to unfamiliar settings and clinical fields in which they are not experts, as well as limited resources, have caused negative physical and mental consequences. Further, exposure to a virus that can endanger life, lack of effective treatments, longer shifts, disturbed sleep patterns, lack of communication or isolation from family, and social stigmatization have contributed to these impacts. Therefore, the care and monitoring of the wellbeing of health professionals became imperative [1,2,3].

In countries such as Singapore, hospitals responded to the pandemic by increasing measures to prevent contagion, detect infection early, reduce care for other pathologies, and cancel elective procedures [4].

In Mexico, tertiary hospitals were converted to hospitals for COVID care, which involved restructuring the facilities as well as changing care services, for which the available staff were given training. They later provided care for COVID-19 patients from triage to critical patient care [5]. In addition, the Mexican army in collaboration with the government of Mexico opened and provided care in new care facilities for patients with COVID-19, and the roles of healthcare professionals changed. Some were displaced to other work facilities, and even to other states in the country.

Working conditions can have a positive or negative impact on the health of employees, the latter of which leads to undesirable consequences for workers, their families, and the work environment [6].

Traumatic events, adverse conditions during natural disasters, conflicts or pandemics can lead to the development of burnout syndrome (BOS) [2].

BOS is defined as a psychological state of physical and mental fatigue associated with work, characterized by three dimensions: (1) emotional exhaustion, which includes emotional exhaustion and loss of energy; (2) depersonalization or cynicism, described as dehumanization, detachment from work and clients and emotional hardening; and (3) reduction of personal accomplishment or ineffectiveness, feelings of personal or professional insufficiency and reduced productivity and coping skills [3,6,7].

BOS effects not only on the psychological and physical state of health workers, such as depression, insomnia and gastrointestinal problems, but also the form of care and the quality of care and satisfaction of staff. This affected labor organization, leading to work absenteeism, bad job satisfaction, repeated rotation, low staff morale and economic losses [3].

The objective of this study is to determine the frequency of BOS and the associated sociodemographic and profession factors in different periods of the COVID-19 pandemic to determine the most effective prevention measures.

## 2. Materials and Methods

Maslach Burnout Inventory self-application was performed in April 2020 (start of the pandemic), June 2020 (first wave peak), September 2020 (end of the first wave) and September 2021 (third wave peak) in a tertiary care hospital in Mexico using the SurveyMonkey^®^ (SurveyMonkey, San Mateo, CA, USA) digital platform (Figure 1). Figure 1 shows the hospital’s number of COVID-19 case admissions per day, reflecting the workload of the hospital.

### 2.1. Ethical Considerations

All subjects gave their digital informed consent for inclusion before they participated in the study. The study was conducted in accordance with the Declaration of Helsinki, and the protocol was approved by the research committee of the Military Central Hospital under number 044/2020. All participants were fully informed that anonymity was assured, and were aware of why the research is being conducted, how their data would be used and if there were any risks associated.

### 2.2. Data Collection

The Maslach Burnout Inventory (MBI) was digitized together with the general sociodemographic data such as age, sex, type of participant (civilian or military), military rank and profession (at the peak of the first wave, undergraduate medicine doctors did not participate, since, by official provisions of the country, they could not be in contact with patients with COVID-19).

The MBI has been validated and accepted internationally and has already been translated into Spanish [8]. The validated instrument was digitized by subject experts; the authors reviewed the questionnaire prior to its dissemination.

At the end of the first evaluation, prevention measures were implemented, including extra hours of rest after the shift, extra days of rest, fewer hours of attention to patients with a diagnosis of COVID-19 with longer periods of rest, rotations of work areas, constant monitoring for development of BOS, virtual sessions on coping strategies and emotional management, orientation to the heads of areas and services for the prevention of burnout and compassion fatigue, military rewards and recognitions and fostering teamwork (esprit de corps).

Because the measures implemented were for all personnel, in this study there was no control group and only follow-up and monitoring were conducted.

The MBI includes three subscales: emotional exhaustion (EE), depersonalization (DP) and personal accomplishment (PA). It consists of 22 questions, evaluated with a Likert-type scale (from 0 = never to 6 = every day).

The MBI evaluation is as shown in the following table.

The definition of BOS requires the presence of the three subscales in red parameters of the semaphore (Table 1).

Once the results of the surveys were obtained, they were reviewed to exclude those that were not answered completely.

### 2.3. Population Characteristics

The participants were invited through official distribution channels of the institution.

The sample calculation was performed for finite populations; the hospital had a staff population of 4100, so the minimum sample was 352.

In all the measurements good participation was obtained; however, some staff were still afraid of mental health stigma, so during the pandemic, campaigns were carried out to promote participation, as well as interventions by the hospital’s mental health staff (psychiatrists and psychologists). These measures, as well as the containment measures implemented in the hospital [9], generated changes in the size of the sample in each of the measurements.

It is important to consider that in the first wave the main changes were related to infrastructure, roles, increases in working hours, scheduled hospital shifts and activities attached to the guidelines established by the Mexican government [5]. However, for the third wave, in addition to the care of patients with COVID-19, elective procedure rooms were reopened, and the hospital was offering participation in vaccination.

### 2.4. Statistical Analysis

The data obtained were analyzed with the IBM SPSS Statistics 25.0 software (SPSS, Inc., Chicago, IL, USA). Descriptive statistics were performed using frequencies and percentages to compare the groups by sex, age, type of participant, military rank, and profession to BOS, and each subscale used X^2^. The profession group was regrouped by type of profession, classifying them as doctors, nurses and others, and reanalyzed with X^2^, considering *p* < 0.05 as statistically significant.

## 3. Results

### 3.1. Sociodemographic Characteristics of the Participants

Most of the participants were female, over 30 years of age, military staff, officers, and general nurses in all the surveys as shown in Table 2. At the peak of the third wave, the participation of civilians and most of the professions was a more homogeneous distribution by profession.

### 3.2. Burnout Syndrome

In total, 2.4% (beginning of the pandemic), 7.9% (peak of the first wave), 3.7% (end of the first wave) and 3.6% (peak of the third wave) of the evaluated personnel presented BOS.

No significant differences were found with respect to age. At the end of the first wave, a higher frequency was observed in males. At the end of the first wave and during the third wave if significant differences were observed between military personnel and civilians, with civilians being the most affected. Significant differences were found only during the first wave in terms of military rank, with bosses being the most affected (Table 3).

At the end of the first wave and at the peak of the third wave, doctors were the most affected; the undergraduate interns and resident doctors were the most affected. Interestingly, the nursing group remained at very similar levels in all measurements, as shown in Table 4.

#### 3.2.1. Affected Subscales

The Maslach scale assesses three items: emotional exhaustion, depersonalization and personal accomplishment.

Most of the participating personnel were in the low range for EE and DP and in the high range for PA, as shown in Figure 2.

At the beginning of the pandemic, the item most affected was PR; in the rest of the evaluations over the course of the pandemic, the item most affected was EE. The highest red levels of EE and DP were at the peak of the first wave, while PA levels were highest at the beginning.

#### 3.2.2. Emotional Exhaustion by Categories Evaluated

Regarding high levels of emotional exhaustion, no significant difference was found with respect to sex in the peak of the first wave. A higher frequency was observed in women compared to men. A significant difference was observed at the end of the first wave and in the peak of the third wave between those older and younger than 30 years. At the peak of the first wave, emotional exhaustion was higher in those over 30 years of age. Civilian personnel showed higher levels at the end of the first wave and during the third wave. Chiefs were the most affected by emotional exhaustion of all military ranks (Table 5).

There were also significant differences in all evaluations by profession. At the beginning, the most affected by emotional exhaustion were the personnel who are not doctors or nurses; however, later, the most affected were the doctors. The undergraduate medicine and resident doctors were the most affected, while the least affected group was the quartermaster staff. Nursing staff remained between 8% and 28%. Specialist physicians went from 10.6% to 23.2%. In all groups, the period with greatest frequency was at the peak of the first wave, as shown in Table 6.

#### 3.2.3. Depersonalization by Categories Evaluated

At the end of the first wave, a higher frequency of high levels of depersonalization was observed in men compared to women. In those under 30 years of age, there was a higher frequency in each of the four measurements; however, it was only significant at the end of the first wave and at the peak of the third wave civilian personnel showed higher levels at the end of the first wave and during the third wave. Only at the peak of the first wave was an increase in the chiefs group observed (Table 7).

Table 8 shows that the groups with the highest levels of depersonalization were physicians from the peak of the first wave, undergraduate interns and resident physicians. Interestingly, for general nursing staff, the frequency levels decreased from 11.7% to 4.8%. The group with the lowest levels of depersonalization was the quartermaster group. The maximum levels of depersonalization in medical residents and specialists were at the peak of the first wave, and these levels decreased at the end of the first wave.

#### 3.2.4. Personal Accomplishment by Categories Evaluated

Table 9 shows that in all the evaluations, there were no significant differences regarding sex and the frequency of low levels of personal accomplishment. However, differences were observed with respect to those under 30 years of age, who experienced low levels of personal accomplishment in all the evaluations except at the peak of the first wave. Civilian personnel showed lower levels at the end of the first wave and during the third wave. At the beginning and at the peak of the third wave, the most affected military rank were chiefs.

At the end of the first wave and during the third wave, doctors were the most affected. The professions most affected were internal physicians, administrative personnel and quartermaster personnel. In the groups of general and specialist nursing, general practitioners, residents and specialists, the frequency of low levels of personal accomplishment decreased with each evaluation (Table 10).

## 4. Discussion

The psychological impact of the COVID-19 pandemic on staff working in hospital centers has previously been studied. Alina Danet’s systematic review shows that these staff had moderate to high levels of stress, anxiety, depression, sleep disturbance and burnout [10]. Diverse coping strategies and more frequent and intense symptoms were more common among women and nurses, without conclusive results by age.

In the present study, the sociodemographic characteristics were similar to those reported in other hospital centers, in which women and nurses predominate [2].

The baseline frequency of BOS (at the beginning of the pandemic) was 2.4%. This was lower than that reported in pre-pandemic periods in the general population, which ranged from 4.3% to 25% [11]. Internal physicians were the most affected, without differences between sex and age.

Hert reported that the most affected group are doctors, determining that in 2020 doctors presented a 43% rate of BOS according to the Medscape National Physician Burnout and Suicide Report. These are similar levels to those reported in 2015 (46%) and in 2013 (39.8%), with women being the most affected. In contrast, among medical staff, we found BOS levels between 2.7% and 2.3% at the beginning of the pandemic; undergraduate internal doctors had BOS levels of 23.5%. We did not find an association with sex, except at the end of the first wave when the most affected group was men [11].

Compared with the first wave, the BOS levels in the third wave decreased. This may be due to the increase in information for diagnosis and treatment, since for this wave, results of various clinical trials were available for the use of treatments such as dexamethasone, tocilizumab and antivirals such as remdesivir. Further, treatment guidelines [12,13,14] as well knowledge about factors of poor prognosis and pathogenesis that allowed better selection of patients who required hospitalization were available, allowing better management of human and economic resources [15]. By this time, vaccination was also available. Vaccination of HCM health personnel began in December 2020 [16], which reduced the uncertainty of the staff; however, fear persisted due to the appearance of new variants against which the effectiveness of the vaccines was unknown [17]. In addition, the burden of work increased due to the fact that procedures unrelated to COVID-19 were carried out almost normally, including elective procedures, and the work roles were now more varied and included providing vaccination support; these factors were reflected in the EE and PA levels.

The prevention measures taken were within the recommendations established in previous pandemics and described in other studies [9]. Since after the peak of the first wave, BOS levels decreased and remained in the third wave, although they did not return to the levels of the beginning of the pandemic. Previously, it has been observed that teamwork, the decision to work, feeling appreciated at work, self-efficacy and emotional support have been associated with less stress, anxiety and burnout. These standards, along with the promotion of military values and esprit de corps and recognition of the personnel who work in the hospital, were the premise of the current study [18].

This esprit de corps and the discipline to which the military is subjected may be the reason why at the end of the pandemic and in the third wave, the military was less affected by BOS in each of its subclasses compared to civilians.

Within the military, the most affected by BOS, EE, and DP were chiefs, especially in the first wave, since they were the ones who had the greatest responsibility. They are also the personnel with the most experience performing administrative and coordination roles. Prior to the pandemic, they did not perform shifts and had a fixed schedule, and focused solely on their medical specialty. Due to the pandemic, they had to attend to COVID patients, from triage to critical patient care, and their schedules were modified, including performing shifts. During the first peak of the pandemic, the highest levels of BOS were observed (7.9%), without significant differences between gender and age. Doctors, rather than nurses or other hospital staff, were the most affected group, which is in contrast to what was found in a multicenter study in Singapore. This study found a BOS rate of 24%, which is higher than that found in our hospital; in addition, they found that the most affected group were nurses [18]. The Singaporean study’s results were similar to those from a study of first-line staff from Japan, who also show that the most affected group was nurses (46.8%). However, among doctors, levels similar to our study (13.4%) were observed. In the case of our study, which showed that the group most affected were doctors. This may be due to greater cognitive, emotional and sensory demands of their role and their great responsibility for the health and lives of others, as well as the need to make difficult decisions quickly and the overload of obligations and concerns [19]. Further, doctors may require a higher level of concentration and state of vigilance, and have greater exposure to suffering and death, which worsens the psychological state of all hospital professionals. Other studies present conflicting results; some have shown that nurses present greater anxiety, depression and stress [10].

In this study, BOS levels were lower than those reported in other studies from Australia (30%), Brazil, (21%), Wuhan (13%–39%), Italy (37%, 25%, 15.3%), Spain (41%, 15.2%, 8.4%) and Portugal (53%), and similar to those reported in Egypt [3]. This suggests that the prevention measures taken by hospital authorities in the present study had a notable impact on workers in general. However, elevated BOS levels compared to baseline may also be related to fear and uncertainty about pathogenesis, lack of treatment guidelines and conflicting initial results on the effectiveness of treatments.

A study conducted in Buenos Aires found that personnel including doctors, nurses, administrative and technicians reported 12% of BOS, with 38%, 33% and 36% of EE, DP, and PA, respectively. In contrast, the present study found lower levels of BOS, similar levels of EE (32.9%) but lower levels of DP (14.9) and PA (19.4), [8].

In Egypt during June–July 2020 (peak of the first wave), it was found that 6% of medial personnel were experiencing BOS, with 35.5%, 70.6% and 26.4% experiencing EE, DP and PA respectively. In our group, between 15.9% and 17.9% experienced BOS. Further, 52.2–47.7%, 34.3–23.4% and 26.9–22.4% presented EE, DP and PA, respectively. Although in our group there were more EE and similar levels of PA, the DP was lower. Resident physicians were more affected than assigned physicians and nurses [3].

The subscale of EE and DP had the highest indices at the peak of the first wave, though they later decreased. The PA levels at the beginning of the pandemic were high but gradually decreased. The low PA levels in our study may be related to greater feelings of achievement when seeing the direct results of treatments in patients with COVID-19, a decrease of cases and deaths, a greater sense of control of the situation, as well as being closer to those who make key decisions. These causes were demonstrated when comparing the survey results of front-line personnel and those who did not care for patients with COVID-19 [20].

In a study by Elghazally et al., doctors who worked more than 8 h had higher levels of EE and PD compared with those who worked less than 4 h or between 4 and 8 h. This is in agreement with our results, since resident doctors worked 24 h shifts every third day, while nurses worked 8 h shifts, in addition to the fact that on some occasions the doctors worked longer days [3]. In addition, in the study by Kannampalli et al., training physicians such as resident physicians presented higher levels of burnout and stress, probably due to modifications in their training and schedules, additional remote work, and an increase in patient care demands. However, the decrease in burnout levels and each of the subscales suggests that the measures implemented in the hospital were efficient (including feeding, availability of personal protective equipment and psychosocial and mental health support). The aforementioned study theorized that clear communication and the feeling of control are protective factors that mitigate adverse effects on mental health [21].

In the study period, we found significant differences between men and women and younger than 30 years for EE; however, we did not find significant differences in PD or AP. This is in contrast to what was found in Egypt where AP was more affected in men [3].

A study carried out by Luceño-Moreno in personnel who were in contact with patients with COVID-19, they found that 41% experienced EE, 15.2% experienced PD and 8.4% experienced AP. Therefore, EE levels were higher in our study, but PD and BP were lower. In this study, they found significant differences in PD between men and women [22].

Regarding the group of interns in Spain, Macía-Rodríguez et al. reported that 40.1% experienced BOS, 58.3% experienced CE, 61.5% experienced PD and 67.6% experienced low PR during the first wave of COVID-19 [23]. Each of these items recorded lower levels than our hospital.

In Italy, they found that the main affected subscale was EE, which is a similar result to our study. They recommend paying attention to this area since it has been associated with suboptimal attention to patients and professional inefficiencies. Further, they theorized that levels of high BP could be a protective factor in the mental health of staff [24].

Most studies on BOS during the COVID-19 pandemic have focused on medical or nursing staff, especially those on the front line of care [1,9,20]. However, when working in the same hospital, multidisciplinary work is necessary, especially in situations like pandemics. Further, BOS is not exclusive to this group of professionals. Therefore, the present study showed how care and surveillance of all personnel is necessary for adequate decision-making and the containment of pathologies associated with work.

## 5. Conclusions

In this study, it was observed that burnout syndrome (BOS) increased primarily due to the increase in EE levels, coinciding with the maximum peak of care in patients. The most affected demographics were men, those under 30 years of age and civilian personnel. Military personnel being less affected than civilians suggests that their military training and esprit de corps give them greater emotional resources to face changing and stressful situations, since during their military service they are accustomed to this type of change. For example, when responding to crisis situations, the Mexican army implements the DN-III-E Plan, whose best known examples are the earthquakes of 1985 and 2017, situations for which civilian personnel are not prepared. On the other hand, although the experience of chiefs allows for better decision-making, it also implies greater responsibility. This, coupled with the role changes to which they were subjected during the first wave of the COVID-19 pandemic, may have influenced their significant increase in BOS, although their personal accomplishment (PA) may also have influenced decrease of BOS and better adaptation at the end of the first wave and in the third wave. On the other hand, although doctors were the most affected group, for personnel in training such as interns and resident doctors, BOS decreased by almost half in the all evaluations after the first, probably due to the decrease in cases and the containment measures carried out in the hospital environment.

## Figures and Tables

**Figure 1 ijerph-19-02229-f001:**
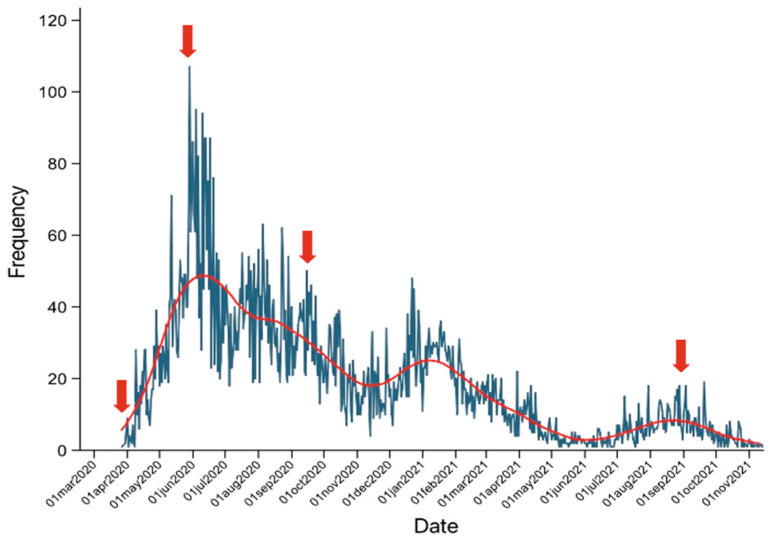
MBI application period with respect to hospital admissions (red arrows).

**Figure 2 ijerph-19-02229-f002:**
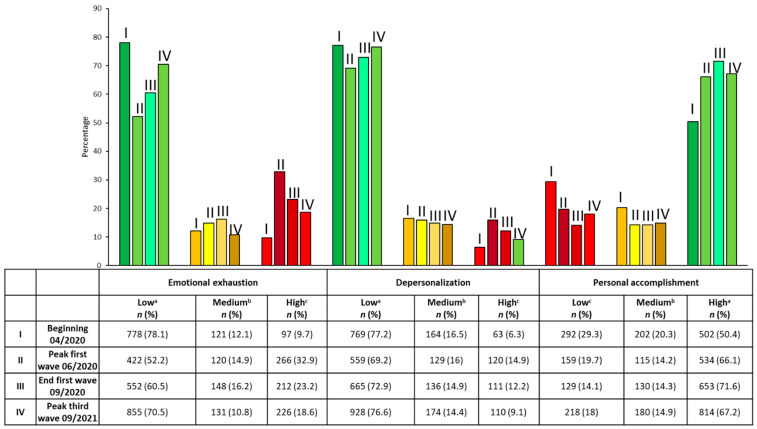
The frequencies and percentages of the levels of each subscale and in traffic light (^a^ green, indicates low risk, ^b^ yellow, indicates intermediate risk, and ^c^ red, indicates burnout) are shown.

**Table 1 ijerph-19-02229-t001:** MBI evaluation.

Subscale	Low	Moderate	High
Emotional exhaustion	0–18 ^a^	19–26 ^b^	27–54 ^c^
Depersonalization	0–5 ^a^	6–9 ^b^	10–30 ^c^
Personal accomplishment	0–33 ^a^	34–39 ^b^	40–56 ^c^

^a^ green color, ^b^ yellow color, ^c^ red color

**Table 2 ijerph-19-02229-t002:** Sociodemographic characteristics and profession of the participants.

Characteristic	Beginning	Peak First Wave	End First Wave	Peak Third Wave
** *N* **	996	808	912	1212
	*n* (%)	*n* (%)	*n* (%)	*n* (%)
**Sex**				
Women	673 (67.6)	549 (67.9)	622 (68.2)	791 (65.3)
Man	323 (32.4)	259 (32.1)	290 (31.8)	421 (34.7)
**Age Range (Years)**				
<30	349 (35)	327 (40.5)	434 (47.6)	487 (40.2)
>30	647 (65)	481 (59.5)	478 (52.4)	725 (59.8)
**Participant Type**				
Civil	17 (1.7)	91 (11.3)	151 (16.6)	251 (20.7)
Military	979 (98.3)	717 (88.6)	761 (83.4)	961 (79.3)
**Military Rank**				
Soldier	252 (25.3)	195 (24.1)	152 (16.7)	249 (20.5)
Class	164 (16.5)	121 (15)	110 (12.1)	280 (23.1)
Official	359 (36)	261 (32.3)	346 (37.9)	306 (25.2)
Chief	204 (20.5)	140 (17.3)	153 (16.8)	126 (10.4)
**Profession**				
Medical specialist	132 (13.3)	107 (13.2)	140 (15.4)	99 (8.2)
General physician	5 (0.5)	1 (0.1)	27 (3)	35 (2.9)
Resident doctor	111 (11.1)	67 (8.3)	16 (1.8)	99 (8.2)
Undergraduate interns	17 (1.7)	0 (0)	163 (17.9)	126 (10.4)
Nurse specialist	94 (9.4)	74 (9.1)	65 (7.1)	78 (6.4)
General nurse	428 (43)	411 (50.8)	416 (45.6)	330 (27.2)
Administrative staff	73 (7.3)	42 (5.2)	30 (3.3)	127 (10.5)
Quartermaster staff	13 (1.3)	12 (1.5)	2 (0.2)	116 (9.6)
Other	123 (12.3)	94 (11.6)	53 (5.8)	202 (16.7)

**Table 3 ijerph-19-02229-t003:** Burnout syndrome.

Pandemic Period	Characteristic	*n* (%)	X^2^, *p*
**Sex**			
**Beginning (4/2020)**	Woman	15 (2.2)	0.289, 0.591
	Man	9 (2.8)	
**Peak first wave (6/2020)**	Woman	37 (6.7)	3.277, 0.070
	Man	27 (10.4)	
**End first wave (9/2020)**	Woman	17 (2.7)	**5.93, 0.02**
	Man	17 (5.9)	
**Peak third wave (9/2021)**	Woman	29 (3.7)	0.008, 0.927
	Man	15 (2.6)	
**Age Group (Years)**			
**Beginning (4/2020)**	<30	8 (2.3)	0.031, 0.859
	>30	16 (2.5)	
**Peak first wave (6/2020)**	<30	22 (6.7)	1.072, 0.301
	>30	42 (8.7)	
**End first wave (9/2020)**	<30	21 (4.8)	2.84, 0.092
	>30	13 (2.7)	
**Peak third wave (9/2021)**	<30	22 (4.5)	1.83, 0.176
	>30	22 (3)	
**Participant Type**			
**Beginning (4/2020)**	Civil	0 (0)	0.513, 0.427
	Military	24 (2.5)	
**Peak first wave (6/2020)**	Civil	4 (4.4)	1.75, 0.186
	Military	60 (8.4)	
**End first wave (9/2020)**	Civil	11 (7.3)	**6.38, 0.012**
	Military	23 (3)	
**Peak third wave (9/2021)**	Civil	24 (9.6)	**31.83, 0.0001**
	Military	20 (2.1)	
**Military Rank**			
**Beginning (4/2020)**	Soldier	4 (1.6)	2.245, 0.523
	Class	3 (1.8)	
	Official	12 (3.3)	
	Chief	5 (2.5)	
**Peak first wave (6/2020)**	Soldier	11 (5.6)	**35.47, 0.0001**
	Class	4 (3.3)	
	Official	16 (6.1)	
	Chief	29 (20.7)	
**End first wave (9/2020)**	Soldier	3 (2)	6.54, 0.088
	Class	1 (0.9)	
	Official	10 (2.9)	
	Chief	9 (5.9)	
**Peak third wave (9/2021)**	Soldier	4 (1.6)	2.62, 0.454
	Class	5 (1.8)	
	Official	6 (2)	
	Chief	5 (4)	

Bold numbers *p* < 0.05.

**Table 4 ijerph-19-02229-t004:** Burnout syndrome by profession.

Pandemic Period	Profession	*n* (%)	X^2^, *p*
**Professions Grouped**			
**Beginning (4/2020)**	Physician	10 (3.8)	3.14, 0.208
	Nurse	9 (1.7)	
	Other	5 (2.4)	
**Peak first wave (6/2020)**	Physician	17 (15.7)	**10.45, 0.005**
	Nurse	37 (6.7)	
	Other	10 (6.8)	
**End first wave (9/2020)**	Physician	26 (7.5)	**22.29, 0.0001**
	Nurse	7 (1.5)	
	Other	1 (1.2)	
**Peak third wave (9/2021)**	Physician	30 (8.4)	**32.58, 0.0001**
	Nurse	7 (1.7)	
	Other	7 (1.6)	
**Professions**			
**Beginning (4/2020)**	Medical specialist	3 (2.3)	**33.89, 0.0001**
	General physician	0 (0)	
	Specialty resident doctor	3 (2.7)	
	Undergraduate medicine doctor	4 (23.5)	
	Nurse specialist	2 (2.1)	
	General nurse	7 (1.6)	
	Administrative staff	2 (2.7)	
	Quartermaster staff	0 (0)	
	Other	3 (2.4)	
**Peak first wave (6/2020)**	Medical specialist	17 (15.9)	**24.757, 0.001**
	General physician	0 (0)	
	Specialty resident doctor	12 (17.9)	
	Undergraduate medicine doctor	NA	
	Nurse specialist	4 (5.4)	
	General nurse	21 (5.1)	
	Administrative staff	3 (7.1)	
	Quartermaster staff	0 (0)	
	Other	7 (7.4)	
**End first wave (9/2020)**	Medical specialist	5 (3.6)	**44.12, 0.0001**
	General physician	3 (11.1)	
	Specialty resident doctor	4 (25)	
	Undergraduate medicine doctor	14 (8.6)	
	Nurse specialist	0 (0)	
	General nurse	7 (1.7)	
	Administrative staff	0 (0)	
	Quartermaster staff	0 (0)	
	Other	1 (1.9)	
**Peak third wave (9/2021)**	Medical specialist	0 (0)	**70.64, 0.0001**
	Specialty resident doctor	11 (11.1)	
	General physician	1 (2.9	
	Undergraduate medicine doctor	18 (14.3)	
	Nurse specialist	2 (2.6)	
	General nurse	5 (1.5)	
	Administrative staff	2 (1.6)	
	Quartermaster staff	1 (0.9)	
	Other	4 (2)	

Bold numbers *p* < 0.05.

**Table 5 ijerph-19-02229-t005:** Emotional exhaustion.

Pandemic Period	Characteristic	Low *n* (%) ^a^	Medium *n* (%) ^b^	High *n* (%) ^c^	X^2^, *p*
**Sex**					
**Beginning (4/2020)**	Woman	522 (77.6)	80 (11.9)	71 (10.5)	1.6, 0.45
	Man	256 (79.3)	41 (12.7)	26 (8)	
**Peak first wave (6/2020)**	Woman	272 (49.5)	91 (16.6)	186 (33.9)	**6.27, 0.044**
	Man	150 (57.9)	29 (11.2)	80 (30.9)	
**End first wave (9/2020)**	Woman	386 (62.1)	102 (16.4)	134 (21.5)	3.23, 0.199
	Man	166 (57.2)	46 (15.9)	78 (26.9)	
**Peak third wave (9/2021)**	Woman	551 (69.7)	87 (11)	153 (19.3)	0.921, 0.631
	Man	304 (72.2)	44 (10.5)	73 (17.3)	
**Age Group (Years)**					
**Beginning (4/2020)**	<30	265 (75.9)	48 (13.8)	36 (10.3)	1.65, 0.438
	>30	513 (79.3)	73 (11.3)	61 (9.4)	
**Peak first wave (6/2020)**	<30	188 (57.5)	42 (12.8)	97 (29.7)	**6.18, 0.046**
	>30	234 (48.6)	78 (16.2)	169 (35.1)	
**End first wave (9/2020)**	<30	232 (53.5)	88 (20.3)	114 (26.3)	**18.45, 0.0001**
	>30	320 (66.9)	60 (12.6)	98 (20.5)	
**Peak third wave (9/2021)**	<30	314 (64.5)	59 (12.2)	114 (23.4)	**15.44, 0.0001**
	>30	541 (74.6)	72 (9.9)	112 (15.4)	
**Participant Type**					
**Beginning (4/2020)**	Civil	14 (82.4)	3 (17.6)	0 (0)	0.341, 2.155
	Military	764 (78)	118 (12.1)	97 (9.9)	
**Peak first wave (6/2020)**	Civil	51 (56)	13 (14.3)	27 (29.7)	0.637, 0.727
	Military	371 (51.7)	107 (14.9)	239 (33.3)	
**End first wave (9/2020)**	Civil	52 (34.4)	40 (26.5)	59 (39.1)	**51.59, 0.0001**
	Military	500 (65.7)	108 (14.2)	153 (20.1)	
**Peak third wave 9/2021)**	Civil	108 (43)	39 (15.5)	104 (41.4)	**128.68, 0.0001**
	Military	747 (77.7)	92 (9.6)	122 (12.7)	
**Military Rank**					
**Beginning (4/2020)**	Soldier	190 (75.4)	40 (15.9)	22 (8.7)	**16.04, 0.014**
	Class	144 (87.8)	10 (6.1)	10 (6.1)	
	Official	279 (77.7)	39 (10.9)	41 (11.4)	
	Chief	151 (74)	29 (14.2)	24 (11.8)	
**Peak first wave (6/2020)**	Soldier	117 (60)	25 (12.8)	53 (27.2)	**31.677, 0.0001**
	Class	76 (62.8)	13 (10.7)	32 (26.4)	
	Official	129 (49.4)	47 (18)	85 (32.6)	
	Chief	49 (35)	22 (15.7)	69 (49.3)	
**End first wave (9/2020)**	Soldier	104 (68.4)	27 (17.8)	21 (13.8)	**30.62, 0.0001**
	Class	89 (80.9)	12 (10.9)	9 (8.2)	
	Official	218 (63)	53 (15.3)	75 (21.7)	
	Chief	89 (58.2)	16 (10.5)	48 (31.4)	
**Peak third wave (9/2021)**	Soldier	206 (82.7)	26 (10.4)	17 (6.8)	**55.99, 0.0001**
	Class	240 (85.7)	15 (5.4)	25 (8.9)	
	Official	230 (75.2)	32 (10.5)	44 (14.4)	
	Chief	71 (56.3)	19 (15.1)	36 (28.6)	

Bold numbers *p* < 0.05; ^a^ green, ^b^ yellow, ^c^ red.

**Table 6 ijerph-19-02229-t006:** Emotional exhaustion by profession.

Pandemic Period	Profession	Low *n* (%) ^a^	Medium *n* (%) ^b^	High *n* (%) ^c^	X^2^, *p*
**Professions Grouped**					
**Beginning (4/2020)**	Physician	190 (71.1)	41 (15.5)	34 (12.8)	**9.46, 0.5**
	Nurse	419 (80.3)	60 (11.5)	43 (8.2)	
	Other	169 (21.7)	20 (9.6)	20 (20.6)	
**Peak first wave (6/2020)**	Physician	39 (36.1)	18 (16.7)	51 (47.2)	**17.52, 0.002**
	Nurse	295 (53.4)	87 (15.8)	170 (30.8)	
	Other	88 (59.5)	15 (10.1)	45 (30.4)	
**End first wave (9/2020)**	Physician	147 (42.5)	69 (19.9)	130 (37.6)	**88.15, 0.0001**
	Nurse	336 (69.9)	72 (15)	73 (15.2)	
	Other	69 (81.2)	7 (8.2)	9 (10.6)	
**Peak third wave (9/2021)**	Physician	149 (41.5)	61 (17)	149 (41.5)	**222.57, 0.0001**
	Nurse	334 (81.9)	37 (9.1)	37 (9.1)	
	Other	372 (83.6)	33 (7.4)	40 (9)	
**Professions**					
**Beginning (4/2020)**	Medical specialist	106 (80.3)	12 (9.1)	14 (10.6)	**40.95, 0.001**
	General physician	2 (40)	3 (60)	0 (0)	
	Specialty resident doctor	73 (65.8)	23 (20.7)	15 (13.5)	
	Undergraduate medicine doctor	9 (52.9)	3 (17.6)	5 (29.4)	
	Nurse specialist	76 (80.9)	11 (11.7)	7 (7.4)	
	General nurse	343 (80.1)	49 (11.4)	36(8.4)	
	Administrative staff	63 (86.3)	6 (8.2)	4 (13)	
	Quartermaster staff	13 (100)	0 (0)	0 (0)	
	Other	93 (75.6)	11 (11.4)	16 (13)	
**Peak first wave (6/2020)**	Medical specialist	38 (35.5)	18 (16.8)	51 (47.7)	**45.77, 0.0001**
	General physician	1 (100)	0 (0)	0 (0)	
	Specialty resident doctor	23 (34.3)	9 (13.4)	35 (52.2)	
	Undergraduate medicine doctor	NA	NA	NA	
	Nurse specialist	44 (59.5)	10 (13.5)	20 (27)	
	General nurse	228 (55.5)	68 (16.5)	115 (28)	
	Administrative staff	23 (54.8)	7 (16.7)	12 (28.6)	
	Quartermaster staff	4 (33.3)	0 (0)	12 (28.6)	
	Other	61 (64.9)	8 (8.5)	25 (26.6)	
**End first wave (9/2020)**	Medical specialist	45 (69.2)	8 (12.3)	12 (18.5)	**137.4, 0.0001**
	General physician	8 (29.6)	7 (25.9)	12 (44.4)	
	Specialty resident doctor	7 (43.8)	1 (6.3)	8 (50)	
	Undergraduate medicine doctor	44 (27)	45 (27.6)	74 (45.4)	
	Nurse specialist	45 (62.9)	8 (12.3)	12 (18.5)	
	General nurse	291 (70)	64 (15.4)	61 (14.7)	
	Administrative staff	24 (80)	2 (6.7)	4 (13.3)	
	Quartermaster Staff	1 (50)	1(50)	0 (0)	
	Other	44 (83)	4 (7.5)	5 (9.4)	
**Peak third wave (9/2021)**	Medical specialist	61 (61.6)	15 (15.2)	23 (23.2)	**306.95, 0.0001**
	General physician	23 (65.7)	3 (8.6)	9 (25.7)	
	Specialty resident doctor	44 (44.4)	17 (17.2)	38 (38.4)	
	Undergraduate medicine doctor	21 (16.7)	26 (20.6)	79 (62.7)	
	Nurse specialist	61 (78.2)	7 (9)	10 (12.8)	
	General nurse	273 (82.7)	30 (9.1)	27 (8.2)	
	Administrative staff	113 (89)	4 (3.1)	10 (7.9)	
	Quartermaster staff	100 (86.2)	7 (6)	9 (7.8)	
	Other	159 (78.7)	22 (10.9)	21 (10.4)	

Bold numbers *p* < 0.05; ^a^ green, ^b^ yellow, ^c^ red.

**Table 7 ijerph-19-02229-t007:** Depersonalization.

Pandemic Period	Characteristic	Low *n* (%) ^a^	Medium *n* (%) ^b^	High *n* (%) ^c^	X^2^, *p*
**Sex**					
**Beginning (4/2020)**	Woman	527 (74.9)	109 (16.2)	37 (5.5)	2.66, 0.264
	Man	242 (74.9)	55 (17)	26 (8)	
**Peak first wave (6/2020)**	Woman	381 (69.4)	96 (17.5)	72 (13.1)	5.97, 0.05
	Man	178 (68.7)	33 (12.7)	48 (18.5)	
**End first wave (9/2020)**	Woman	469 (75.4)	92 (14.8)	61 (9.8)	**10.66, 0.005**
	Man	196 (67.6)	44 (15.2)	50 (17.2)	
**Peak third wave (9/2021)**	Woman	608 (76.9)	111 (14)	72 (9.1)	0.194, 0.908
	Man	320 (76.0)	63 (15)	38 (9)	
**Age Group (Years)**					
**Beginning (4/2020)**	<30	267 (76.5)	59 (16.9)	23 (6.6)	0.16, 0.924
	>30	502 (77.6)	105 (16.2)	40 (6.2)	
**Peak first wave (6/2020)**	<30	220 (67.3)	57 (17.4)	50 (15.3)	1.1, 0.577
	>30	339 (70.5)	72 (15)	70 (14.6)	
**End first wave (9/2020)**	<30	294 (67.7)	73 (16.8)	67 (15.4)	**12.32, 0.002**
	>30	371 (77.6)	63 (13.2)	44 (9.2)	
**Peak third wave (9/2021)**	<30	337 (69.2)	90 (18.5)	60 (12.3)	**24.86, 0.0001**
	>30	591 (81.5)	84 (11.6)	50 (6.9)	
**Participant Type**					
**Beginning (4/2020)**	Civil	14 (82.4)	2 (11.8)	1 (5.9)	0.297, 0.862
	Military	755 (77.1)	162 (16.5)	62 (6.3)	
**Peak first wave (6/2020)**	Civil	64 (70.3)	18 (19.8)	9 (9.9)	2.654, 0.265
	Military	495 (69)	111 (15.5)	111 (15.5)	
**End first wave (9/2020)**	Civil	85 (56.3)	34 (22.5)	32 (21.2)	**25.98, 0.0001**
	Military	580 (76.2)	102 (13.4)	79 (10.4)	
**Peak third wave (9/2021)**	Civil	146 (58.2)	54 (21.5)	51 (20.3)	**69.38, 0.0001**
	Military	782 (81.4)	120 (12.5)	59 (6.1)	
**Military Rank**					
**Beginning (4/2020)**	Soldier	185 (73.4)	49 (19.4)	18 (7.1)	5.721, 0.455
	Class	126 (76.8)	31 (18.9)	7 (4.3)	
	Official	286 (79.7)	51 (14.2)	22 (6.1)	
	Chief	158 (77.5)	31 (15.2)	15 (7.4)	
**Peak first wave (6/2020)**	Soldier	131 (67.2)	37 (19)	27 (13.8)	**45.38, 0.0001**
	Class	89 (73.6)	27 (22.3)	5 (4.1)	
	Official	188 (72)	37 (14.2)	36 (13.8)	
	Chief	87 (62.1)	10 (7.1)	43 (30.7)	
**End first wave (9/2020)**	Soldier	112 (73.3)	24 (15.8)	16 (10.5)	9.53, 0.146
	Class	89 (89.9)	16 (14.5)	5 (4.5)	
	Official	264 (76.3)	47 (13.6)	35 (10.1)	
	Chief	115 (75.2)	15 (9.8)	23 (15)	
**Peak third wave (9/2021)**	Soldier	199 (79.9)	35 (14.1)	15 (6)	12.07, 0.06
	Class	237 (84.6)	31 (11.1)	12 (4.3)	
	Official	254 (83)	35 (11.5)	17 (5.6)	
	Chief	92 (73)	19 (15.1)	15 (11.9)	

Bold numbers *p* < 0.05; ^a^ green, ^b^ yellow, ^c^ red.

**Table 8 ijerph-19-02229-t008:** Depersonalization by profession.

Pandemic Period	Profession	Low *n* (%) ^a^	Medium *n* (%) ^b^	High *n* (%) ^c^	X^2^, *p*
**Professions Grouped**					
**Beginning (4/2020)**	Physician	198 (74.7)	42 (15.8)	25 (9.4)	8.371, 0.079
	Nurse	414 (79.3)	80 (15.3)	28 (5.4)	
	Other	769 (77.2)	164 (16.5)	63 (6.3)	
**Peak first wave (6/2020)**	Physician	75 (69.4)	8 (7.4)	25 (23.1)	**12.19, 0.016**
	Nurse	380 (68.8)	94 (17)	78 (14.1)	
	Other	104 (70.3)	27 (18.2)	17 (11.5)	
**End first wave (9/2020)**	Physician	228 (65.9)	55 (15.9)	63 (18.2)	**22.34, 0.0001**
	Nurse	370 (76.9)	67 (13.9)	44 (9.1)	
	Other	67 (78.8)	14 (16.5)	4 (4.7)	
**Peak third wave (9/2021)**	Physician	216 (60.2)	72 (20.1)	71 (19.8)	**94.23, 0.0001**
	Nurse	337 (82.6)	50 (12.3)	21 (5.1)	
	Other	375 (84.3)	52 (11.7)	18 (4)	
**Profession**					
**Beginning (4/2020)**	Medical specialist	108 (81.8)	16 (12.1)	8 (6.1)	**34.19, 0.005**
	General physician	4 (80)	1 (20.0)	0 (0)	
	Specialty resident doctor	74 (66.7)	24 (21.6)	13 (11.7)	
	Undergraduate medicine doctor	12 (70.6)	1 (5.9)	4 (23.5)	
	Nurse specialist	80 (85.1)	12 (12.8)	2 (2.1)	
	General nurse	334 (78)	24 (21.6)	13 (11.7)	
	Administrative staff	57 (78.1)	12 (16.4)	4 (5.5)	
	Quartermaster staff	7 (53.8)	6 (46.2)	0 (0)	
	Other	93 (75.6)	24 (19.5)	6 (4.9)	
**Peak first wave (6/2020)**	Medical specialist	74 (69.2)	8 (7.5)	25 (23.4)	**44.32, 0.0001**
	General physician	1 (100)	0 (0)	0 (0)	
	Specialty resident doctor	35 (52.2)	9 (13.4)	23 (34.3)	
	Undergraduate medicine doctor	NA	NA	NA	
	Nurse specialist	59 (79.7)	8 (10.8)	25 (23.4)	
	General nurse	286 (69.6)	77 (18.7)	48 (11.7)	
	Administrative staff	26 (61.9)	9 (21.4)	7 (16.7)	
	Quartermaster staff	9 (75)	3 (25)	0 (0)	
	Other	69 (73.4)	15 (16)	10 (10.6)	
**End first wave (9/2020)**	Medical specialist	108 (77.1)	15 (10.7)	17 (12.1)	**52.05, 0.0001**
	General physician	14 (51.9)	5 (18.5)	8 (29.6)	
	Specialty resident doctor	8 (50)	3 (18.8)	5 (31.3)	
	Undergraduate medicine doctor	98 (60.1)	32 (19.6)	33 (20.2)	
	Nurse specialist	108 (77.1)	15 (10.7)	17 (12.1)	
	General nurse	313 (75.2)	61 (14.7)	42 (10.1)	
	Administrative staff	28 (93.3)	2 (6.7)	0 (0)	
	Quartermaster staff	1 (50)	1(50)	0 (0)	
	Other	38 (71.7)	11 (20.8)	4 (7.5)	
**Peak third wave (9/2021)**	Medical specialist	80 (80.8)	14 (14.4)	5 (5.1)	**178.14, 0.0001**
	General physician	26 (74.3)	6 (17.1)	3 (8.6)	
	Specialty resident doctor	61 (61.6)	13 (13.1)	25 (25.3)	
	Undergraduate medicine doctor	49 (38.9)	39 (31)	38 (30.2)	
	Nurse specialist	67 (85.9)	6 (7.7)	5 (6.4)	
	General nurse	270 (81.8)	44 (13.3)	16 (4.8)	
	Administrative staff	103 (81.1)	19 (15)	5 (3.9)	
	Quartermaster staff	106 (91.4)	7 (6)	3 (2.6)	
	Other	166 (82.2)	26 (12.9)	10 (5)	

Bold numbers *p* < 0.05; ^a^ green, ^b^ yellow, ^c^ red.

**Table 9 ijerph-19-02229-t009:** Personal accomplishment.

Pandemic Period	Characteristic	Low *n* (%) ^c^	Medium *n* (%) ^b^	High *n* (%) ^a^	X^2^, *p*
**Sex**					
**Beginning (4/2020)**	Woman	208 (30.9)	139 (20.7)	326 (48.4)	3.51, 0.173
	Man	84 (26)	63 (19.5)	176 (54.4)	
**Peak first wave (6/2020)**	Woman	101 (18.4)	85 (15.5)	363 (66.1)	3.31, 0.191
	Man	58 (22.4)	30 (11.6)	171 (66)	
**End first wave (9/2020)**	Woman	84 (13.5)	86 (13.8)	452 (72.2)	1.13, 0.569
	Man	45 (15.5)	44 (15.2)	201 (69.3)	
**Peak third wave (9/2021)**	Woman	147 (18.6)	123 (15.5)	521 (65.9)	1.77, 0.413
	Man	71 (16.9)	57 (13.5)	293 (69.6)	
**Age Group (Years)**					
**Beginning (4/2020)**	<30	132 (37.8)	61 (17.5)	156 (44.7)	**18.8, 0.0001**
	>30	160 (24.7)	141 (21.8)	346 (53.5)	
**Peak first wave (6/2020)**	<30	72 (22)	43 (13.1)	212 (64.8)	2.11, 0.348
	>30	87 (18.1)	72 (15)	322 (66.9)	
**End first wave (9/2020)**	<30	75 (17.3)	76 (17.5)	283 (65.2)	**16.65, 0.0001**
	>30	54 (11.3)	54 (11.3)	370 (77.4)	
**Peak third wave (9/2021)**	<30	93 (19.1)	90 (18.5)	304 (62.4)	**10.49, 0.005**
	>30	125 (17.2)	90 (12.4)	510 (70.3)	
**Participant Type**					
**Beginning (4/2020)**	Civil	1 (5.9)	6 (35.3)	10 (58.8)	5.41, 0.067
	Military	291 (29.7)	196 (20)	492 (50.3)	
**Peak first wave (6/2020)**	Civil	13 (14.3)	15 (16.5)	63 (69.2)	2.03, 0.362
	Military	146 (20.4)	100 (13.9)	471 (65.7)	
**End first wave (9/2020)**	Civil	29 (19.2)	35 (23.2)	87 (57.6)	**18.33, 0.0001**
	Military	100 (13.1)	95 (12.5)	566 (74.4)	
**Peak third wave (9/2021)**	Civil	56 (22.3)	56 (22.3)	139 (55.4)	**21.69, 0.0001**
	Military	162 (16.9)	124 (12.9)	675 (70.2)	
**MilitaryRank**					
**Beginning (4/2020)**	Soldier	101 (40.1)	42 (16.7)	109 (43.3)	**21.10, 0.002**
	Class	43 (26.2)	30 (18.3)	91 (55.5)	
	Official	101 (28.1)	80 (22.3)	178 (49.6)	
	Chief	46 (22.5)	44 (21.6)	114 (55.9)	
**Peak first wave (6/2020)**	Soldier	49 (25.1)	25 (12.8)	121 (62.1)	11.86, 0.065
	Class	20 (16.5)	16 (13.2)	85 (70.2)	
	Official	40 (15.3)	38 (14.6)	183 (70.1)	
	Chief	37 (26.4)	21 (15)	82 (58.6)	
**End first wave (9/2020)**	Soldier	30 (19.7)	17 (11.2)	105 (69.1)	7.584, 0.270
	Class	14 (12.7)	13 (11.8)	83 (75.5)	
	Official	38 (11)	45 (13)	263 (76)	
	Chief	18 (11.8)	20 (13.1)	115 (75.2)	
**Peak third wave (9/2021)**	Soldier	52 (20.9)	34 (13.7)	163 (65.5)	**18.47, 0.005**
	Class	58 (20.7)	42 (15)	180 (64.3)	
	Official	38 (12.4)	36 (11.8)	232 (75.8)	
	Chief	14 (11.1)	12 (9.5)	100 (79.4)	

Bold numbers *p* < 0.05; ^a^ green, ^b^ yellow, ^c^ red.

**Table 10 ijerph-19-02229-t010:** Personal accomplishment by profession.

Pandemic Period	Profession	Low *n* (%) ^c^	Medium *n* (%) ^b^	High *n* (%) ^a^	X^2^, *p*
**Professions Grouped**					
**Beginning (4/2020)**	Physician	64 (24.2)	53 (20)	148 (55.8)	6.655, 0.155
	Nurse	161 (30.8)	102 (19.5)	259 (49.6)	
	Other	67 (32.1)	47 (22.5)	95 (45.5)	
**Peak first wave (6/2020)**	Physician	24 (22.2)	16 (14.8)	68 (63)	0.772, 0.942
	Nurse	105 (19)	79 (14.3)	368 (66.7)	
	Other	30 (20.3)	20 (13.5)	98 (66.2)	
**End first wave (9/2020)**	Physician	57 (16.5)	65 (18.8)	224 (64.7)	**14.33, 0.006**
	Nurse	62 (12.9)	57 (11.9)	362 (75.3)	
	Other	10 (11.8)	8 (9.4)	67 (78.8)	
**Peak third wave (9/2021)**	Physician	73 (20.3)	69 (19.2)	217 (60.4)	**20.39, 0.0001**
	Nurse	55 (13.5)	48 (11.8)	305 (74.8)	
	Other	90 (20.2)	63 (14.2)	292 (65.6)	
**Profession**					
**Beginning (4/2020)**	Medical specialist	23 (17.4)	29 (22)	80 (60.6)	**31.41. 0.012**
	General physician	1 (20)	0 (0)	4 (80)	
	Resident doctor	32 (28.8)	22 (19.8)	57 (51.4)	
	Undergraduate interns	8 (47.5)	2 (11.8)	7 (41.2)	
	Nurse specialist	17 (18.1)	17 (18.1)	60 (63.8)	
	General nurse	144 (33.6)	85 (19.9)	199 (46.5)	
	Administrative staff	26 (35.6)	14 (19.2)	33 (45.2)	
	Quartermaster staff	2 (15.4)	5 (38.5)	6 (46.2)	
	Other	39 (31.7)	28 (22.8)	56 (45.5)	
**Peak first wave (6/2020)**	Medical specialist	24 (22.4)	16 (15)	67 (62.6)	18.34, 0.192
	General physician	0 (0)	0 (0)	1 (100)	
	Resident doctor	18 (26.9)	14 (20.9)	35 (52.2)	
	Undergraduate interns	NA	NA	NA	
	Nurse specialist	11 (14.9)	8 (10.8)	55 (74.3)	
	General nurse	76 (18.5)	57 (13.9)	278 (67.6)	
	Administrative staff	12 (28.6)	3 (7.1)	27 (64.3)	
	Quartermaster staff	0 (0)	4 (33.3)	8 (66.7	
	Other	18 (19.1)	13 (13.8)	63 (67)	
**End first wave (9/2020)**	Medical specialist	15 (10.7)	16 (11.4)	109 (77.9)	**51.1, 0.0001**
	General physician	5 (18.5)	7 (25.9)	15 (55.6)	
	Resident doctor	4 (25)	3 (18.8)	9 (56.3)	
	Undergraduate interns	33 (20.2)	39 (23.9)	91 (55.8)	
	Nurse specialist	15 (10.7)	16 (11.4)	109 (77.9)	
	General nurse	56 (13.5)	52 (12.5)	308 (74)	
	Administrative staff	3 (10)	4 (13.3)	23 (76.7)	
	Quartermaster staff	2 (100)	0 (0)	0 (0)	
	Other	5 (9.4)	4 (7.5)	44 (83)	
**Peak third wave (9/2021)**	Medical specialist	8 (8.1)	10 (10.1)	81 (81.8)	**86.73, 0.0001**
	General physician	2 (5.7)	5 (14.3)	28 (80)	
	Resident doctor	21 (21.2)	20 (20.2)	58 (58.6)	
	Undergraduate interns	42 (33.3)	34 (27)	50 (39.7)	
	Nurse specialist	9 (11.5)	7 (9)	62 (79.5)	
	General nurse	46 (13.9)	41 (12.4)	243 (73.6)	
	Administrative staff	21 (16.5)	19 (15)	87 (68.5)	
	Quartermaster staff	36 (31)	16 (13.8)	64 (55.2)	
	Other	33 (16.3)	28 (13.9)	141 (69.8)	

Bold numbers *p* < 0.05; ^a^ green, ^b^ yellow, ^c^ red.

## Data Availability

The data that support the findings of this study are available on request from the corresponding author.

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
