# Peer review of "Burnout Syndrome in a Military Tertiary Hospital Staff during the COVID-19 Contingency"

_ijerph, 2022, doi:10.3390/ijerph19042229_

Round 1
Reviewer 1 Report
The introduction does not provide sufficient background about the tertiary hospital and its situation during each stage of the COVID-19 contingency - it could be varied between countries. The structure of the method and discussion section could be improved. Authors should describe the implications more, as research has an important application potential.
Author Response
The introduction does not provide sufficient background about the tertiary hospital and its situation during each stage of the COVID-19 contingency - it could be varied between countries.
Thank you, we added the information on the main text as follow:
As in other countries such as Singapore, hospitals responded to the pandemic by increasing measures to prevent contagion, early detection, reducing care for other pathologies, and canceling elective procedures [1].
In Mexico, tertiary hospitals were converted to hospitals for COVID care, which involved restructuring the facilities as well as changing care services, for which the available staff was given training and later dedicated themselves to caring for patients with COVID-19, from COVID triage to critical patient care [2]. In addition, the Mexican Army in collaboration with the Government of Mexico opened and provided care in new care facilities for patients with COVID, that in addition to the change in the role of the professional, some were displaced to other work facilities, even in other states from the country.
- Cai, Y.; Kwek, S.; Tang, S.S.L.; Saffari, S.E.; Lum, E.; Yoon, S., Ansah, J.P.; Matchar, D.B.; Kwa, A.L.; Ang, K.A.; Thumboo, J.; Ong, M.E.H.; Graves, N. Impact of the COVID-19 pandemic on a tertiary care public hospital in Singapore: resources and economic costs. J Hosp Infect. 2021, 11(121), 1-8. doi: 10.1016/j.jhin.2021.12.007.
- Lineamientos-Reconversion-Hospitalaria. Gobierno de México | Secretaría de Salud. COVID-19. Available online: https://coronavirus.gob.mx/wp-content/uploads/2020/04/Documentos-Lineamientos-Reconversion-Hospitalaria.pdf. (accessed on 04 February 2022).
The structure of the method and discussion section could be improved
Thank you, we reordered the method as follow:
2.3. Population characteristics
The participants were invited through official distribution channels of the institution. The sample calculation was performed for finite populations, the hospital has 4100 people, so the minimum sample is 352.
In all the measurements good participation was obtained, however, some people from the staff were still afraid of mental health stigmas, so during the pandemic campaigns were carried out to promote participation, as well as interventions by the hospital´s mental health staff (psychiatrists & psychologists), among the containment measures implemented in the hospital [3], which generated changes in the size of the sample in each of the measurements.
It is important to consider that in the first wave the main changes were related to infrastructure changes, role changes, increase in working hours, scheduled hospital shifts, as well as activities attached to the guidelines established by the Mexican government [4]. However, for the third wave, in addition to the care of patients with COVID-19, elective procedure rooms were reopened, and the hospital was already offering participation in vaccination.
- Restauri, N.; Sheridan, A.D. Burnout and Posttraumatic Stress Disorder in the Coronavirus Disease 2019 (COVID-19) Pandemic: Intersection, Impact, and Interventions. J Am Coll Radiol. 2020, Jul 1;17(7):921–6.
- Lineamientos-Reconversion-Hospitalaria. Gobierno de México | Secretaría de Salud. COVID-19. Available online: https://coronavirus.gob.mx/wp-content/uploads/2020/04/Documentos-Lineamientos-Reconversion-Hospitalaria.pdf. (accessed on 04 February 2022).
2.4. Statistical analysis
The data obtained were analyzed with the IBM SPSS Statistics 25.0 software. The descriptive statistics were performed, using frequencies and percentages, as well as inferential to compare the groups by sex, age, type of participant, military rank, and profession with the BOS, and each subscale was used statistics using X2, in the case of profession, it was regrouped by type of profession, classifying them as doctors, nurses and others, and reanalyzed with the X2, considering p <0.05 as statistically significant.
About the discussion section it was improved on the main text
Authors should describe the implications more, as research has an important application potential.
Thank you for your comment, we have added the next paragraph:
Most studies on BOS during the COVID-19 pandemic focus on medical and/or nursing staff, especially those on the front line of care [5,6,7], however, when working in the same hospital, interpersonal relationships can affect others and multidisciplinary work is necessary, especially in situations like this pandemic, added to the fact that BOS is not exclusive to this group of professionals, so this study shows how care and surveillance of all personnel is necessary for adequate decision-making and the containment of pathologies associated with work.
- Raudenská, J.; Steinerová, V.; Javůrková, A.; Urits, I.; Kaye, A.D.; Viswanath, O.; Varrassi, G. Occupational burnout syndrome and post-traumatic stress among healthcare professionals during the novel coronavirus disease 2019 (COVID-19) pandemic. Best Pract Res Clin Anaesthesiol. 2020, 34(3), 553–60.
- Restauri, N.; Sheridan, A.D. Burnout and Posttraumatic Stress Disorder in the Coronavirus Disease 2019 (COVID-19) Pandemic: Intersection, Impact, and Interventions. J Am Coll Radiol. 2020, Jul 1;17(7):921–6.
- Wu, Y.; Wang, J.; Luo, C.; Hu, S.; Lin, X.; Anderson, A.E.; Bruera, E.; Yang, X.; Wei, S.; Qian, Y. A Comparison of Burnout Frequency Among Oncology Physicians and Nurses Working on the Frontline and Usual Wards During the COVID-19 Epidemic in Wuhan, China. J Pain Symptom Manage. 2020, 60(1), e60-e65. doi: 10.1016/j.jpainsymman.2020.04.008.
Reviewer 2 Report
Thank you for the opportunity to review this original article. The universal phenomenon of BOS among healthcare workers during the ongoing pandemic is a topic of significant interest and is also an issue of concern in order to protect the healthcare workforce.
This study reflects the situation in Argentina and the authors are to be commended for presenting comparisons with other countries.
The lower level of BOS during the third wave may be due to increased clinical and epidemiological knowledge of COVID-19 and the availability of more effective treatment methods. Perhaps this aspect can be mentioned in Discussion.
I note one typo in line 49: it should read "can endanger".
Author Response
Thank you for the opportunity to review this original article. The universal phenomenon of BOS among healthcare workers during the ongoing pandemic is a topic of significant interest and is also an issue of concern in order to protect the healthcare workforce.
This study reflects the situation in Argentina and the authors are to be commended for presenting comparisons with other countries.
The lower level of BOS during the third wave may be due to increased clinical and epidemiological knowledge of COVID-19 and the availability of more effective treatment methods. Perhaps this aspect can be mentioned in Discussion.
Thank you, we have added information about your suggestions
Compared with the first wave, the BOS levels in the third wave decreased, this may be due to the increase in information on diagnosis and treatment, since for this wave, there were results on various clinical trials such as the use of dexamethasone, tocilizumab, antivirals such as remdesivir, and treatment guidelines were available [1–3]; as well as knowledge about factors of poor prognosis and pathogenesis that allowed better selection of patients who required hospitalization, allowing better management of human and economic resources [4], added to vaccination; in the Military Central Hospital the staff vaccination began in December 2020 [5], which reduced the uncertainty of people, however, fear persisted due to the appearance of new variants of which the effectiveness of the vaccines was unknown [16] and the burden of work increased due to the fact that the rest of the procedures were carried out almost normally, including elective procedures, and the work roles were now mixed, including the support given in the vaccination process; what is reflected in the EE and the PA.
- Guideline Clinical management of COVID-19 patients: living guideline, 18 November 2021. 2021. Available online: https://www.who.int/publications/i/item/WHO-2019-nCoV-clinical-2021-2. (accessed on 04 February 2022).
- Mouffak, S.; Shubbar, Q.; Saleh, E.; El-Awady, R. Recent advances in management of COVID-19: A review. Biomed Pharmacother. 2021, 143, 112107. doi: 10.1016/j.biopha.2021.112107
- Mishra, S.K.; Tripathi, T. One year update on the COVID-19 pandemic: Where are we now? Acta Trop. 2021, 214, 105778. doi: 10.1016/j.actatropica.2020.105778.
- Kim, H.J.; Hwang, H.; Hong, H.; Yim, J.J.; Lee, J. A systematic review and meta-analysis of regional risk factors for critical outcomes of COVID-19 during early phase of the pandemic. Sci Rep. 2021 7;11(1), 9784. doi: 10.1038/s41598-021-89182-8.
- García-Araiza, M.G.;Martinez-Cuazitl, A.; Cid-Dominguez, B.E.; Uribe-Nieto, R.; Ortega-Portillo, R.; Almeyda-Farfán, J.A.; Barrón-Campos, A.C.; Martínez-Salazar, I.N. Side effects of the BNT162b2 vaccine in the personnel of the Military Central Hospital. Eur Rev Med Pharmacol Sci. 2021, 25(19), 5942-5946. doi: 10.26355/eurrev_202110_26871.
- Bian, L.; Gao, F.; Zhang, J.; He, Q.; Mao, Q.; Xu, M.; Liang, Z. Effects of SARS-CoV-2 variants on vaccine efficacy and response strategies. Expert Rev Vaccines. 2021, 20(4), 365-373. doi: 10.1080/14760584.2021.1903879.
I note one typo in line 49: it should read "can endanger".
Thank you, it has been corrected
Reviewer 3 Report
- The abstract lacks the characteristics of the research group.
- The introduction is complete and clear.
- Materials are described very generally. It would be worthwhile to provide a more detailed description of the research tool used.
- The results obtained are described well.
- Table 4, Table 5 and Table 6 are connected with each other, which makes them difficult to understand. I propose to separate them more clearly, it is worth adding a short description under each of the tables.
- The discussion is interesting and well written. It would be good to point out the practical implications.
Author Response
The abstract lacks the characteristics of the research group.
Thank you, we have added the lack information as follow:
Methods: The Maslach Inventory (MBI) was self-applied in 4 periods of the pandemic and the sociodemographic and employment characteristics; in this study, all hospital personnel were included, the association of BOS with sex, age, type of participant (civilian or military), military rank and profession were analyzed.
The introduction is complete and clear.
Materials are described very generally. It would be worthwhile to provide a more detailed description of the research tool used.
Thank you, the materials section was reorganized
The results obtained are described well.
Table 4, Table 5 and Table 6 are connected with each other, which makes them difficult to understand. I propose to separate them more clearly, it is worth adding a short description under each of the tables.
Thank you, we have reordered the tables.
The discussion is interesting and well written. It would be good to point out the practical implications.
Thank you, we have added information about the topic on the main text.
Most studies on BOS during the COVID-19 pandemic focus on medical and/or nursing staff, especially those on the front line of care [1,9,25], however, when working In the same hospital, interpersonal relationships can affect others and multidisciplinary work is necessary, especially in situations like this pandemic, added to the fact that BOS is not exclusive to this group of professionals, so this study shows how care and surveillance of all personnel is necessary for adequate decision-making and the containment of pathologies associated with work
Reviewer 4 Report
It is a long work in time and number of participants and also in a complicated period which is to be appreciated that despite everything they have carried out the study
I'm going to comment on a few points.
It would be advisable not to use the same keywords as those used in the title. When searching, information is lost. If you can put keywords different from those of the title, if possible
Line 100 replace Burnout syndrome with BSO. And in successive places where it appears (line 112, etc)
Statistical analysis should be a bit broader, making a better description of the statistical methods used.
Table 2 introduces UI: undergraduate interns. I don't quite understand why it is set when not in use
In table 3. Analyzing the professions in this way by having such a low N the values ​​of X2 and p are not well resolved. In some cases, you have an n of zero. I think it is more convenient that the separation by professions is not made to see if there is significance.
Something similar happens with table 4
It is quoted: "Curiously, in the general nursing staff, the frequency levels decreased from 11.7% to 4.8%". Can you explain this statement in some way?
In lines 229 and 272 you present the abbreviation SBO. It has not been referenced
Another piece of information that leaves me in doubt is that the participants are not the same so at the beginning of the pandemic you have 996 individuals and at the end, you have 1,212. You are comparing different individuals who may have evolved psychologically differently, and even have participants who start to participate or work at the end of the third wave. If this is the case, they may or may not have developed BSO. As it has been controlled that in the peak 3rd wave those who participate are not new personnel who have not yet developed BSO. This point has to be clarified in material and methods.
Apart from the length of the work and the amount of data and results that you obtain very few conclusions from the work. Conclusions can be drawn on whether women present more or less BSO if more in civilians or military if the military rank affects etc.
I think that with the data you can squeeze a little more and draw more precise conclusions and not just a general one. The article will have more impact if you get these
Author Response
It is a long work in time and number of participants and also in a complicated period which is to be appreciated that despite everything they have carried out the study
I'm going to comment on a few points.
It would be advisable not to use the same keywords as those used in the title. When searching, information is lost. If you can put keywords different from those of the title, if possible
Thank you, we change some key words: burnout syndrome; COVID-19 pandemic; tertiary hospital; military and civilian personnel.
Line 100 replace Burnout syndrome with BSO. And in successive places where it appears (line 112, etc)
Thank you, it was corrected.
Statistical analysis should be a bit broader, making a better description of the statistical methods used.
Thank you, we have added some information on this section as follow:
2.4. Statistical analysis
The data obtained were analyzed with the IBM SPSS Statistics 25.0 software. The descriptive statistics were performed, using frequencies and percentages, as well as inferential to compare the groups by sex, age, type of participant, military rank, and profession with the BOS, and each subscale was used statistics using X2, in the case of profession, it was regrouped by type of profession, classifying them as doctors, nurses and others, and reanalyzed with the X2, considering p <0.05 as statistically significant.
Table 2 introduces UI: undergraduate interns. I don't quite understand why it is set when not in use
Thank you, it was eliminated.
In table 3. Analyzing the professions in this way by having such a low N the values ​​of X2 and p are not well resolved. In some cases, you have an n of zero. I think it is more convenient that the separation by professions is not made to see if there is significance.
Something similar happens with table 4
Thank you, we have reordered all the tables, regarding your valuable suggestions about the others analysis.
It is quoted: "Curiously, in the general nursing staff, the frequency levels decreased from 11.7% to 4.8%". Can you explain this statement in some way?
It is interesting, that general nurses since the beginning of the pandemic and in the first wave remained in the same levels, nevertheless, at the end of the first wave it decreased fast, and continued in the same level in the third wave, but the rest of the groups presented and increment and gradually deacreased.
In lines 229 and 272 you present the abbreviation SBO. It has not been referenced
Thank you, we have correct that abbreviation.
Another piece of information that leaves me in doubt is that the participants are not the same so at the beginning of the pandemic you have 996 individuals and at the end, you have 1,212. You are comparing different individuals who may have evolved psychologically differently, and even have participants who start to participate or work at the end of the third wave. If this is the case, they may or may not have developed BSO. As it has been controlled that in the peak 3rd wave those who participate are not new personnel who have not yet developed BSO. This point has to be clarified in material and methods.
Thank you, we have added this information on the methodology section as follow:
The participants were invited through official distribution channels of the institution.
The sample calculation was performed for finite populations, the hospital has 4100, so the minimum sample is 352.
In all the measurements good participation was obtained, however, some people from the staff were still afraid of mental health stigmas, so during the pandemic campaigns were carried out to promote participation, as well as interventions by the hospital´s mental health staff (psychiatrists & psychologists), among the containment measures implemented in the hospital [9], which generated changes in the size of the sample in each of the measurements.
It is important to consider that in the first wave the main changes were related to infrastructure changes, role changes, increase in working hours, scheduled hospital shifts, as well as activities attached to the guidelines established by the Mexican government [5]. However, for the third wave, in addition to the care of patients with COVID-19, elective procedure rooms were reopened, and the hospital was already offering participation in vaccination.
Apart from the length of the work and the amount of data and results that you obtain very few conclusions from the work. Conclusions can be drawn on whether women present more or less BSO if more in civilians or military if the military rank affects etc.
I think that with the data you can squeeze a little more and draw more precise conclusions and not just a general one. The article will have more impact if you get these
Thank you so much, we have improved the conclusion section as follow:
In this study, it was observed that BOS increased mainly due to the increase in EE levels, coinciding with the maximum peak of care in patients, the most affected being men, under 30 years of age, civilian personnel, being a minor in military personnel suggesting that their training and esprit de corps give them greater emotional resources to face changing and stressful situations, since during their military service they are accustomed to this type of changes, for example its response to crisis situations, considering that the Mexican Army has the DN-III-E PLAN which is activated in the face of this type of crisis, whose best known examples are the earthquakes of 1985 and 2017, situations for which the civilian personnel is not prepared, on the other hand, although the experience of the chiefs personnel allows for better decision-making, it also implies greater responsibility coupled with the role changes to which they were subjected during the first wave could be the factors that influenced such a significant increase in BOS, although its PA could influence the decrease and better adaptation at the end of the first wave and in the third wave, on the other hand, although the doctors were the most affected, the personnel in training such as interns and resident doctors, however, this decreased almost by half in the following evaluation periods, probably due to the decrease in cases, but also due to the containment measures carried out in the hospital environment.
Round 2
Reviewer 4 Report
The changes made after the revision better clarify the work and leave no questions.